# Knowledge, Attitudes, and Practices Related to COVID-19 Among Malawi Adults: A Community-Based Survey

**DOI:** 10.3390/ijerph18084090

**Published:** 2021-04-13

**Authors:** Yutong Li, Guangqi Liu, Robert Okia Egolet, Runqing Yang, Yangmu Huang, Zhijie Zheng

**Affiliations:** 1Department of Global Health, School of Public Health, Peking University, 38 Xue Yuan Road, Haidian District, Beijing 100191, China; lyt1997wang@163.com (Y.L.); liugq@pku.edu.cn (G.L.); zhengzj@bjmu.edu.cn (Z.Z.); 2Institute for Global Health, Peking University, Beijing 100191, China; 3Global Health Collaborating Centre for Research and Training in Health Sciences, Peking University, P.O. Box 166, Lilongwe 265, Malawi; egoletroberts@gmail.com; 4Department of Economics, HSBC Business School, Peking University, University Town, Nanshan District, Shenzhen 518055, China; runqyang@pku.edu.cn

**Keywords:** KAP, challenge, infodemic, LMIC, COVID-19, Malawi

## Abstract

Introduction: It is well-recognized that containing COVID-19 successfully is determined by people’s prevention measures which are related to their knowledge, attitudes, and practices (KAP). This perception has attracted attention in low- and middle-income countries (LMIC) due to their fragile health systems and economies. The objective of this study was to understand how residents in Malawi perceived COVID-19, to determine the factors related to KAP. Methods: A semi-structured questionnaire was used for the data collection. A field-based survey was conducted among adult residents in Lilongwe, Malawi. Descriptive statistic, linear regression, the Chi-square test, and Pearson’s correlation statistics were used for data analysis. Results: A total of 580 questionnaires were involved. The mean knowledge, attitude, and practice (KAP) scores were 10 (SD = ±3, range: 3–19), 16 (SD = ±4, range: 5–25), and 2 (SD = ±1, range: 0–5), respectively. Lack of money and resources (39%) was the biggest challenge for people who practice prevention measures. Among the participants, the radio (70%) and friends/family (56%) were the main sources of information. A higher economic status was associated with better KAP. Conclusions: A low level of KAP was detected among the population. The people faced challenges regarding a lack of necessary preventive resources and formal information channels. The situation was worse considering vulnerable population who had low economic status. Further all-round health education is urgently needed along with providing adequate health supplies and ensuring proper information management.

## 1. Introduction

COVID-19 is a respiratory infection caused by the Severe Acute Respiratory Syndrome Coronavirus 2 (SARS-CoV-2) [1], which the World Health Organization (WHO) declared a global pandemic in March 2020 [2]. As of 1 March 2021, COVID-19 has caused nearly 113 million confirmed cases and over 2.5 million deaths [3].

COVID-19 had severe consequences both in developing countries and developed countries. Low- and middle-income countries (LMIC) face a more significant crisis considering the resource gap and pressures on health care systems [4]. Also, infodemic such as stigmatization have hampered the positive response to COVID-19 [5], affecting public preventive behaviors against COVID-19. A clear understanding of the public perception towards COVID-19 in LMIC is needed.

The knowledge, attitude, and practice (KAP) model is widely used in the medical field. The model suggests that practices (behaviors) are determined by the person’s attitude and knowledge [6]. Assessing KAP in terms of COVID-19 would help us to understand public perception and their response to COVID-19, which is related to public adherence to the preventive measures [7]. There have been some KAP surveys conducted in LMIC. Several surveys (conducted in Iran, Syria, and Bangladesh) revealed a low or middle level of KAP among the public [8,9,10,11], indicating the enormous challenges faced by LMIC for containing COVID-19. Some of the surveys in LMIC were online surveys, in which the electronic questionnaire usually could not cover the vulnerable populations in LMIC since they have less access to the Internet. Thus, further surveys based on field interviews are important for a better understanding of COVID-19 in LMIC.

Malawi is one of the world’s poorest countries, with a population of 18.6 million (2019) [12]. Health care funding in Malawi is low, heavily relying on donor resources, and an effective and efficient delivery of health care services is lacking [13]. Malawi registered the first cases of COVID-19 on 1 April 2020 and it was declared a national disaster immediately [14]. The government took measures to contain the virus. These included border control, closing schools, urging people of avoid public activities, and suspending visas [15]. By 31 March 2021, COVID-19 caused 33,535 (prevalence: 0.2%) cases and 1116 (case fatality rate: 3.3%) deaths [16], with the fatality rate in Malawi being higher than the global case fatality rate of 2.9% [17]. Lilongwe, the capital city of Malawi, is also the transportation center of Malawi. As of 31 March 2021, there were 7590 cases and 272 deaths in Lilongwe [16], accounting for nearly 23% of the cases within Malawi. 

To improve the prevention management of Malawi, there is an urgent need to know how the residents perceive the disease. The objective of this study was to understand the knowledge, attitude, and practice (KAP) of the residents regarding COVID-19 in Lilongwe of Malawi, and determine the factors related to KAP towards COVID-19.

## 2. Methods

### 2.1. Study Design

A cross-sectional design was used for this survey. The fieldwork was conducted in Lilongwe, Malawi by Peking University Research and Training Centre in Malawi (PKURTC) from 31 August to 7 September 2020.

The target population was Malawi residents at the age of 18 years or more living in Lilongwe. Participants who had difficulties in communication such as listening and speaking or who were not willing to participate were excluded from the study.

Considering a population of one million (2015) in Lilongwe, a sample size of 385 was recommended with an assumption of a 95% confidence interval (CI) regarding a 5% margin of error and a response rate of 50%. Considering various errors in finishing the questionnaire, the research team added a further 50% (*N* = 192) to the sample, resulting in a final sample size of 577.

The study adopted a two-stage sampling technique consisting of the selection of residential areas and individuals. For the primary sampling unit, the study used simple cluster sampling based on the list of Lilongwe’s administrative divisions, and several areas were selected. Within each selected area, the amounts of sample were population weighted. The systematic sampling of households was done according to house numbers, and household heads were included in the KAP survey.

A semi-structured questionnaire was used for the data collection. The questionnaire was digitalized and programmed on tablets using Open Data Kit (ODK) software, version 1.28.4. Investigators were assigned to each area and captured individual-level quantifiable indicators face to face. The questionnaire was premeditated according to the WHO Zika KAP Resource Pack [18] and some KAP surveys regarding COVID-19 conducted in other countries [19,20,21]. The research team conducted the pre-test, and it needed about 14 min to finish the questionnaire. At last, the questionnaire was modified following the suggestion of some experts. It was initially prepared in English and then translated into Chichewa (see online Appendix A).

The questionnaire consisted of four sections: (1) Details of demographic characteristics including gender, age, education, residence, occupation, marital status, and economic status. The information sources regarding COVID-19 were also inquired along with two questions regarding stigmatization; (2) 7 items in the knowledge section investigating the participants’ awareness regarding symptoms, asymptomatic infection, transmission, the population at risk, preventive measures, effective cure and antibiotic; (3) 5 items in the attitude section assessing the people’s response to COVID-19; and (4) 4 items in the practice section understanding the prevention measures applied by the participants and the difficulty they faced when practicing COVID-19 prevention measures.

### 2.2. Statistical Analysis

For questions in the knowledge and practice section, a score of 1 was attributed to a correct answer and 0 to a wrong answer or uncertain, respectively. For the attitude section, a five-point Likert Scale was used for the assessment of the attitude score, in which different statements were scored by 1 for “strongly disagree”, 2 for “disagree”, 3 for “neutral”, 4 for “agree”, 5 for “strongly agree”. Economic status was calculated by two questions related to house condition and family properties. The 25%, 50%, and 75% quintiles of its distribution indicated the bounds among high, upper-middle, lower-middle and low economic status. Descriptive statistics (frequency and proportions) were used to present the demographic characteristics of the participants. Pearson’s correlation statistics were used to assess the relationship among KAP. A multivariable linear regression analysis was performed to identify factors related to KAP and find the relationship in each category. The Chi-square test was used to detect the association between information resources and concealment behaviors regarding COVID-19. The statistical significance level was set at *p* < 0.05 (two-sided). Data analysis was conducted using the IBM Statistical Package for social science (SPSS) version 26.0 (IBM, Chicago, IL, USA).

## 3. Results

### 3.1. Demographic Characteristic

A total of 586 questionnaires were received. After removing 6 respondents who answered “no” for COVID-19 awareness, eventually 580 questionnaires were used for final analysis.

Among the participants, 65% were females, 72% married, and 82% were less than 45 years old. Only 4% had an education level of more than secondary while 14% had no education, and most participants (40%) had an incomplete primary education. The majority of the participants (70%) were rural residents. In terms of occupation status, 44% reported relying on agriculture, 23% unemployed, 22% sales and services, 7% manual labor, and 4% professional/technical. In addition, 21% of the participants had high economic status whereas 22% had low economic status.

### 3.2. Knowledge

The mean knowledge score regarding COVID-19 was 10 (SD = ±3, range: 3–19). The overall accuracy rate for the test was 48% (10/21*100). The details are shown in Table 1. Fever (62%) and dry cough (74%) were known as the common symptoms of COVID-19 by over half of the participants. Most participants (76%) realized that close contact could cause the transmission of the virus. Additionally, only 38% of the participants were aware of the asymptomatic infection regarding COVID-19. Considering prevention measures, most participants were aware of the importance of washing hands (79%) and wearing masks (70%). Meanwhile, 61% of the participants knew that there was no special drug or effective treatment for COVID-19 and 68% of the participants did not think antibiotic should be used for treatment.

### 3.3. Attitude

Overall, the mean attitude score concerning COVID-19 was 16 (SD = ±4, range: 5–25). Most participants (65%) thought that COVID-19 was not an important issue/problem in their community. Furthermore, 54% agreed they were concerned about being infected by COVID-19 and 48% attested to the disturbance of COVID-19 in their daily life, respectively. On the other hand, 67% believed that COVID-19 could be controlled successfully while 73% thought that the government’s measures were quite effective. Details of the attitudes section are presented in Table 2.

### 3.4. Practice

The mean score of the practice section was 2 (SD = ±1, range: 0–5), indicating poor practice among the participants. The study acknowledged that 486 (84%) participants had taken part in the prevention activity. Regarding prevention measures available, the two most common measures were washing hands (69%) and wearing masks (75%). A few participants were practicing social distancing (33%) and avoidance of crowded places (46%). Details are presented in Table 3.

From the findings, a good number of participants (39%) claimed that they had difficulties in taking measures against COVID-19 transmission because they lacked money and resources. Meanwhile, 33% mentioned that they had less access to necessary preventive supplies (Figure 1). For the 16% of people who did not adopt any prevention measures, the reasons varied. Most people (61%) did not take any action because of a lack of resources or access, while 11% thought that they and their family were not at risk, and 12% thought COVID-19 was not a problem.

Linear regression showed the factors related to KAP (Table 4). For knowledge regarding COVID-19, people who had low economic status were less knowledgeable than those of a higher economic status. Also, people with higher economic status were better at practices against COVD-19 transmission. People who had no education held a more relaxed attitude than the other group. Regarding occupation, people engaged in agriculture (β = 0.431, *p* < 0.05), manual labor (β = 0.714, *p* < 0.05), sales and services (β = 0.456, *p* < 0.05), and professional/technical/managerial/clerical (β = 1.325, *p* < 0.05) were better at practice than people who were unemployed. In addition to the above, Pearson’s correlation statistics confirmed that there was also a relationship among KAP. The results showed that *r* between KAP were 0.419 (Knowledge and Attitude, *p* < 0.05), 0.711 (Knowledge and Practice, *p* < 0.05) and 0.415 (Attitude and Practice, *p* < 0.05), respectively.

### 3.5. Information

The main source of information was the radio (407, 70%), and over half of the participants claimed they got information from family and friends (326, 56%). In addition, social media (64, 11%) and the internet (54, 9%) were not popular among the participants. Among the participants, 124 (21%) preferred to keep it a secret if somebody in their family were to get COVID-19, and over half of the participants (326, 56%) claimed there was indeed a stigmatization of COVID-19. The Chi-square test showed that there was a relationship between information sources and concealment behaviors. People who got information through newspaper/magazine or leaflet were less likely to conceal COVID-19 status and had better knowledge regarding COVID-19 (Table 5).

## 4. Discussion

COVID-19 is a global pandemic that has resulted in a high death rate and economic crises across the globe. LMIC are facing enormous challenges considering their fragile health systems. Health care facilities in some LMIC were already overcrowded with those suffering from pneumonia, human immunodeficiency virus (HIV), tuberculosis (TB,) and malaria [4]. Also, there continues to be a lack of sanitation services and hygiene facilities [22]. Preventive measures play an important role in containing the pandemic while public adherence to preventive measures is influenced by their knowledge and attitude toward COVID-19. This study revealed the knowledge, attitudes, and practices of residents in Lilongwe, Malawi, concerning COVID-19. The results demonstrated that the population was not knowledgeable about COVID-19 and held poor practices against its transmission. The most important factors related to KAP were education and economic status. People who get a higher level of education held a more cautious attitude. Higher economic status was associated with better knowledge and practice. It suggested that health education around COVID-19 is still needed, and it would be more effective focused towards the population with low economic status. The results were consistent with a survey conducted in Saudi Arabia, which found that less educated and lower-income people were less knowledgeable about COVID-19 [20].

Insufficiency regarding COVID-19 knowledge was detected, which was also attested by another survey in Malawi [23]. Also, a significant knowledge gap was observed regarding treatments of COVID-19. Participants were not clear about the asymptomatic infection of COVID-19. Some participants thought there was effective treatment available for COVID-19 and antibiotics use was agreeable. In addition, there was a great number of participants who answered “Don’t know” for these three questions. Interestingly, some surveys in other LMIC countries (Malaysia, Bangladesh, Egypt, and Nigeria) showed the people had good knowledge of these aspects [7,11,24]. The main reason for this difference might be that the proportion of participants with higher education in these countries’ surveys is higher than that of Malawi. This indicated that the preparedness of the public in Malawi was inadequate. In order to reduce the inappropriate use of drugs, it is necessary to establish proper guidelines on the treatment efficacy and use of antibiotics. It would be more effective to deliver the guidance through healthcare professionals in clinical practice. For example, pharmacists and physicians could ask the patients about the purpose of purchasing the medicine, remind them that these drugs are not used to treat COVID-19, and clarify that preventive measures are most effective for containing COVID-19.

It was noted that many participants did not show enough concern regarding the disease. On the other side, some were confident that the disease would be solved. This result was not similar to the result of a survey conducted in China, in which people held optimistic attitudes about containing COVID-19 while still practiced caution [19]. Another survey in India showed that the people there were concerned about COVID-19 while they were not sure about the government response [25]. This disparity in the results could be attributed to Malawi’s overwhelming disease burden, which reduced the attention regarding COVID-19, and inadequate knowledge reduced Malawians’ sensitivity to the disease. Notably, the relaxed attitude could be a reason for the poor practice, as a number of the people in the survey who did not take prevention practice claimed they thought COVID-19 was not a problem and they were not at risk.

It is more challenging for the vulnerable populations in LMIC to deal with the pandemic, considering the weak health systems and limited resources [26]. The survey found that the biggest challenge for taking measures was the lack of resources and access to essential supplies. In fact, the whole country was caught in a situation of lacking health resources. In the hospital, there have been crucial gaps in resources needed to treat patients with COVID-19 infection, especially the inadequate oxygen in medical wards, resulting in avoidable deaths. For all four central (tertiary) hospitals in Malawi, only one had an outpatient or emergency department while no one had an intensive care unit. The lack of PPE such as masks and gowns also posed a substantial risk to the healthcare workers and the general public [27]. What made the matter worse was that people with low economic status not only were restricted because of lacking resources, they also had less knowledge about COVID-19 than the other people.

The quality of information is essential as the infodemic has arisen and it has potentially severe implications on public health [5]. It is noted that the distortion in the process of information dissemination such as rumors, stigma, and conspiracy theories could help initiate and spread misinformation [28]. Some surveys noticed the identification of information sources. A survey in Cameroon found that three most significant information sources were television, Whatsapp, and websites [29]. The survey in Nigeria also reported that people heard of COVID-19 mostly from the internet, social media, and television [30]. Unlike these countries, people in Malawi have less access to electronic equipment and the Internet which were considered as a favored mechanism for spreading misinformation [31]. Participants claimed their information sources were mainly radio and relatives, which was also consistent with the previous survey in Malawi [23]. Our survey indicated that information sources were different in reliability and may influence on people’s behavior. Further research is needed to identify the difference between information sources.

If people contracting diseases are discriminated against because of misinformation, they will prefer to keep a secret for disease and reduce health-seeking behaviors [32]. People in LMIC already suffer from less accessibility to health services because of overwhelmed health system and inadequate health resources. At the same time, the stigma around COVID-19 will further decrease their willingness to seek health care. The double effects are deemed to increase the distance between the residents and available health care services. In that case, information management needs to be considered to deal with the infodemic. The cooperation among the government, the public, and the media (especially social media platform) is essential to establish a platform that monitors information channels to identify misinformation and respond quickly to provide facts and precautionary messages. Moreover, a recent research suggested that it is possible to practice a kind of “inoculation” for fake news. Just like medical vaccination, the idea is to give a small or weakened dosage of the harmful substance. This type of practice allows the reader or viewer develop immunity before the true threat appears [33].

Our survey may be useful for policymakers and healthcare professionals. The findings indicated the targeted population of health education were people with low-level economic status. There is also a need for aid to provide the necessary preventative gear to low-income communities. Additionally, awareness campaigns against misinformation are needed to ensure access to reliable information issued by health authorities [34]. In summary, further public health intervention should not only focus on providing knowledge of COVID-19, but also changing the public’s attitude towards the disease and encouraging them to be more cautious about it. A long-lasting health education program with wide coverage could be helpful. One way to deliver effective education could be to create a confidential online system to share COVID-19 experiences and consult online. It is worth mentioning that sending medical staff directly to the community may better help vulnerable populations. In addition to the above, our study provides a broader insight for researchers who could use the same research design to collect data under similar circumstances; learning from what is happening in Malawi could be useful for comparative studies on COVID-19 experiences of other African countries.

### Limitation

This study also encountered limitations. Because of travel restriction and time difference, researchers could not follow up on the surveys in time and conduct in-depth interviews for further research such as identifying the real situation of stigmatization towards COVID-19 patients. There was also a limitation regarding the less representativeness for the other Malawi regions, particularly those in hard-to-reach areas. Also, certain characteristics are divided into many different groups while the sample size of each group is relatively small, which leads to higher variability in the data distribution. Further study containing a larger population is needed.

## 5. Conclusions

This study assessed the KAP of the residents in Lilongwe of Malawi. This population is not knowledgeable about COVID-19, holds a relaxed attitude towards it, and has poor prevention practices. For LMIC facing emerging infectious disease, the lack of adequate resources is deemed to be a prevalent problem hampering the practice for prevention measures, and there is less formal channel for high-quality information regarding COVID-19. It is urgently needed to provide all-round health education and essential prevention supplies for the residents and strengthen the formal information channels for reliable information. More emphasis is needed for vulnerable populations with low economic status.

## Figures and Tables

**Figure 1 ijerph-18-04090-f001:**
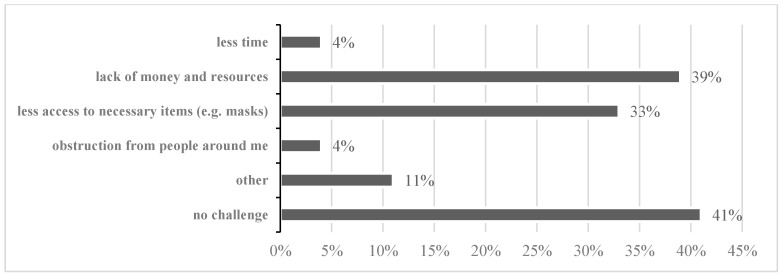
Challenges when practicing prevention measures.

**Table 1 ijerph-18-04090-t001:** Participants’ knowledge about COVID-19.

Knowledge	*N*	%
What are the signs and symptoms of COVID-19?		
Fever	362	62
Fatigue	136	23
Dry cough	430	74
Headache	164	28
Myalgia	214	37
Shortness of breath	274	48
Don’t know	48	8
Does everybody who gets COVID-19 show symptoms?		
Yes	221	38
No	228	39
Don’t know	131	23
How does a person get COVID-19?		
Close contact with someone infected by COVID-19	443	76
Talking with someone coughing and sneezing	287	50
Sexual intercourse	7	1
Mosquito bite	5	1
Eating wild animals	18	3
Touching surface of items attached by virus	161	28
Don’t know	65	11
Who can get COVID-19?		
Aged persons	128	22
Adult men	113	20
Adult women	117	20
Boys	99	17
Girls	96	17
People with chronic diseases (HIV, Diabetes, etc.)	72	12
Health workers	62	11
Everybody can get COVID-19	481	83
Don’t know	25	4
How can people prevent COVID-19?		
Washing hands	460	79
Avoiding touching your eyes, nose, and mouth with unwashed hands	207	36
Use of disinfectants	16	3
Herbal supplements	20	3
Covering your cough	74	13
A balanced diet	11	2
Staying home when you are sick	29	5
Avoiding close contact with someone who is sick	185	32
Use caution when opening mail	4	1
Getting the flu shot	2	0.5
Regular exercise	6	1
Wearing a face mask	407	70
Drinking alcohol	4	1
Don’t know	23	4
Is there a special drug or an effective treatment for COVID-19?		
Yes	57	10
No	352	61
Don’t know	171	29
You should use antibiotic to treat COVID-19.		
Yes	49	8
No	395	68
Don’t know	136	24

**Table 2 ijerph-18-04090-t002:** Attitudes towards COVID-19.

Attitudes	Strongly Disagree *N* (%)	Disagree *N* (%)	Neutral *N* (%)	Agree *N* (%)	Strongly Agree *N* (%)
COVID-19 is an important issue/problem in your community.	93 (16)	285 (49)	23(4)	134 (23)	45 (8)
I am concerned about that I would be infected by COVID-19.	77 (13)	145 (25)	43 (8)	244 (42)	71 (12)
My life has been disturbed by COVID-19.	93 (16)	182 (31)	28 (5)	194 (34)	83 (14)
COVID-19 would be controlled successfully.	15 (3)	73 (12)	103 (18)	270 (47)	119 (20)
The measures of the government are effective.	14 (3)	59 (10)	80 (14)	278 (48)	149 (25)

**Table 3 ijerph-18-04090-t003:** Practices towards COVID-19.

Statement	*N*	%
Have you taken any action to prevent yourself?		
Yes	486	84
No	91	15
Don’t know	3	1
Measures for prevention		
Avoiding going to crowded places.	224	46
Avoiding taking public transportation.	45	9
Washing hands more frequently.	335	69
Wearing masks when leaving home.	362	75
Practicing social distancing.	160	33
Praying to god.	46	10
Don’t know.	4	1

**Table 4 ijerph-18-04090-t004:** Regression results of factors related to the knowledge, attitudes, and practices (KAP) for COVID-19.

Variable	Knowledge		Attitude		Practice	
	β	*p*-Value	β	*p*-Value	β	*p*-Value
Gender (REF: Female)						
Male	0.048	0.856	0.241	0.440	0.032	0.783
Female						
Age (REF: 55+)						
18–25	0.900	0.100	−0.066	0.918	0.070	0.769
26–35	1.462	<0.05	1.000	0.116	0.327	0.167
36–45	0.719	0.190	0.705	0.278	0.177	0.462
46–55	1.542	<0.05	0.618	0.378	0.368	0.158
Residence (REF: Rural)						
Urban	0.207	0.543	0.523	0.194	0.304	<0.05
Education (REF: No education)						
Primary incomplete	0.684	0.074	1.162	<0.05	0.176	0.293
Primary complete	2.864	<0.05	2.122	<0.05	1.112	<0.05
Secondary incomplete	1.276	<0.05	2.369	<0.05	0.522	<0.05
Secondary complete	2.492	<0.05	3.338	<0.05	0.850	<0.05
More than secondary	−0.171	0.851	4.076	<0.05	0.342	0.393
Occupation (REF: Unemployed)						
Agriculture	0.453	0.214	1.160	<0.05	0.431	<0.05
Manual labor	0.924	0.077	0.843	0.172	0.714	<0.05
Sales and services	0.425	0.247	0.728	0.094	0.456	<0.05
Professional/technical/managerial/clerical	2.745	<0.05	0.584	0.576	1.325	<0.05
Marital status (REF: Not Married)						
Married	−0.434	0.240	−0.265	0.544	−0.213	0.189
Divorced/separated	0.038	0.949	−1.053	0.128	−0.158	0.538
Widowed	−0.590	0.425	0.099	0.910	0.265	0.416
Economic status (REF: Low)						
Lower middle	0.919	<0.05	0.043	0.912	0.423	<0.05
Upper middle	2.166	<0.05	0.670	0.118	0.890	<0.05
High	3.275	<0.05	1.119	<0.05	1.260	<0.05

**Table 5 ijerph-18-04090-t005:** The results of the Chi-square test for the association between information sources and concealment behaviors of COVID-19.

Information Sources		If Somebody in My Family Were to Get COVID-19, I Would Want to Remain a Secret		*p*-Value	Knowledge	
		Yes	No		Mean	SD
Television	Yes	15	59	0.740	13	3
	No	109	387			
Radio	Yes	86	316	0.746	11	3
	No	38	130			
Newspaper/magazine	Yes	2	32	<0.05	14	3
	No	122	414			
Social media	Yes	8	55	0.065	12	4
	No	116	391			
Internet	Yes	9	45	0.341	14	3
	No	115	401			
Health care professional	Yes	27	143	<0.05	12	4
	No	97	303			
Friends and family	Yes	60	188	0.215	11	4
	No	64	258			
Leaflet	Yes	3	55	<0.05	16	2
	No	121	391			

## Data Availability

All data generated during this study are included in this published article and its Appendix A.

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
