# Peer review of "Knowledge, Attitudes, and Practices Related to COVID-19 Among Malawi Adults: A Community-Based Survey"

_ijerph, 2021, doi:10.3390/ijerph18084090_

Round 1

Reviewer 1 Report

This paper has potential to make practical contributions. Below, I would like to suggest a few points that might be helpful for future revision.

  1. Line 41: What were the key findings of previous studies on LMIC using survey methods? Please explain. Also, there needs to be theoretical/conceptual explanation of why KAP is important for the success of preventive measures.

  1. Line 51: What is the number of total population in Malawi? It will be helpful to understand the seriousness of the situation if the authors could show the percentage of COVID-19 cases compared to the number of total population.

  1. Line 56: What is the implication of the location ‘Lilongwe’? Is it a rural area with low income? Some description of the study location would be helpful. Also, a detailed description of the COVID-19 situation in the study location (Lilongwe) would be helpful in finding practical implications. For example, how serious is the pandemic situation in the area compared to Malawi as a whole?

  1. Line 78: What does ‘ONA’ stand for?

  1. Line 78: each area and captured  

  1. Line 88: What were the two questions related to stigmatization? What was the purpose of measuring stigmatization? Also, how many items did each KAP categories contain?

  1. Table 1: Further explanation of the findings in Table 1 is required. The significant group differences (e.g., knowledge differs significantly across age groups, etc.) need explanation.

  1. Line 156: From, not Form

  1. Line 166: Linear

  1. Line 180: Please mention Table 6 in the paragraph.

  1. Line 195: Please describe some details about the fragile health systems in LMIC.

  1. Line 213: Who are these people in other surveys? Were these surveys also conducted in Malawi? What may be the reason for the different results between other surveys and this study?

  1. Line 214: What may be an effective way to provide proper guidance? Some practical suggestions would be helpful.

  1. Line 221: could be attributed to

  1. Line 234: less knowledge about

  1. Line 251: Please describe the overwhelmed health system and inadequate health resources in Malawi.

  1. Line 254: What could be done to manage information? Please explain.

  1. Line 261: What could be done to change attitude?

Reviewer 2 Report

Congratulation on your hard work in collating these surveys from a large number of people living in Malawi. It is important that the results of this study are known so more support can be given to the management of chronic disease in struggling economies especially during a pandemic. I have made some editorial suggests below.

  • Page 1, Line 41: Even though you have correctly written the words in full in the abstract and then shown the abbreviations. In the main paper, you need to write the words in full again in the introduction before proceeding e.g. low- and middle-income countries (LMIC) and knowledge, attitude, and practice (KAP)
  • Page 1, Line 43: ‘Thus, further survey based on field interview is important for proper understanding concerning COVID-19 in LMIC’ should read ‘Thus, further surveys based on field interviews are important for a better understanding of COVID-19 in LMIC’
  • Page 2, Line 51: add the percentage for Malawi numbers. ‘31945 cases and 1040 (XX%) deaths’
  • Page 2, line 88: use the word section rather than part.
  • Page 3, line 117: ‘Only 4.1% got an education level of more than secondary’ change to ‘Only 4% had got an education level of more than secondary’
  • Page 3, Line 116 to 120: make all percentages whole numbers e.g. 64.8% should be 65% etc.
  • Table 1, 2, 3, 5 and 6: make all numbers whole numbers including SD except p values
  • Make sure all percentages and SD are whole numbers throughout the manuscript e.g.
    • Page 4, Line 124 to 134: make all numbers whole numbers including SD except p values
    • Page 5, Line 138 to 144: make all percentages and SD whole numbers
  • Figure 1: make results whole numbers
  • You have a lot of tables and I am wondering if you could consolidate the information to reduce the number of Tables. Ideally results should only be presented once – either in a Table or Figure OR in text. Check that you have not repeated yourself.
  • Discussion: Line 208: ‘Insufficiency regrading COVID-19 knowledge was detected..’ should read ‘Insufficiency regarding COVID-19 knowledge was detected…’
  • Line 213: ‘People in other surveys showed clear good knowledge of these aspects’ should read ‘People in other surveys showed clear good knowledge of these aspects’
  • Line 221: ‘It could be attribute to the overwhelming disease burden in Malawi which alleviated the attention regarding COVID-19, and inadequate knowledge reducing the people’s sensitiveness to the disease’. Rewrite as ‘It could be attributed to the overwhelming disease burden in Malawi which reduced alleviated the attention regarding COVID-19, and inadequate knowledge reduced ing the peoples ’s sensitivity eness to the disease.
  • Line 249: ‘If people contacting diseases…’ should be ‘If people contracting diseases…’ check the rest of this sentence
  • In this paragraph there is a variation in font size
  • Conclusion: This study assessed KAP of towards the residents in Lilongwe of Malawi. This population is not knowledgeable about COVID-19, while they hold a relaxed attitude towards the it , and have a poor practice of n prevention.

This is a very interesting study and congratulations on your hard work
